# Combination of PDT and NOPDT with a Tailored BODIPY Derivative

**DOI:** 10.3390/antiox8110531

**Published:** 2019-11-07

**Authors:** Loretta Lazzarato, Elena Gazzano, Marco Blangetti, Aurore Fraix, Federica Sodano, Giulia Maria Picone, Roberta Fruttero, Alberto Gasco, Chiara Riganti, Salvatore Sortino

**Affiliations:** 1Department of Drug Science and Technology, University of Torino, 10125 Torino, Italy; loretta.lazzarato@unito.it (L.L.); federica.sodano@unito.it (F.S.); roberta.fruttero@unito.it (R.F.); alberto.gasco@unito.it (A.G.); 2Department of Oncology, University of Torino, 10126 Torino, Italy; elena.gazzano@unito.it (E.G.); giulia.picone@edu.unito.it (G.M.P.); 3Department of Chemistry, University of Torino, 10125 Torino, Italy; marco.blangetti@unito.it; 4Department of Drug Sciences, University of Catania, 95125 Catania, Italy; fraix@unict.it

**Keywords:** photodynamic therapy, singlet oxygen, nitric oxide, light, combination therapy

## Abstract

The engineering of photosensitizers (PS) for photodynamic therapy (PDT) with nitric oxide (NO) photodonors (NOPD) is broadening the horizons for new and yet to be fully explored unconventional anticancer treatment modalities that are entirely controlled by light stimuli. In this work, we report a tailored boron-dipyrromethene (BODIPY) derivative that acts as a PS and a NOPD simultaneously upon single photon excitation with highly biocompatible green light. The photogeneration of the two key species for PDT and NOPDT, singlet oxygen (^1^O_2_) and NO, has been demonstrated by their direct detection, while the formation of NO is shown not to be dependent on the presence of oxygen. Biological studies carried out using A375 and SKMEL28 cancer cell lines, with the aid of suitable model compounds that are based on the same BODIPY light harvesting core, unambiguously reveal the combined action of ^1^O_2_ and NO in inducing amplified cancer cell mortality exclusively under irradiation with visible green light.

## 1. Introduction

Photodynamic therapy (PDT) for cancer involves the combined use of nontoxic photosensitizers (PSs) and visible light of appropriate wavelength [1,2]. Upon light excitation, PSs reach the lowest excited singlet state (^1^PS*) and, after intersystem crossing (ISC), their lowest-energy excited triplet state (^3^PS*). Due to their long lifetime (µs-ms), the ^3^PS* can dissipate most of their energy via quenching with nearby molecular oxygen, potentially giving rise to two different reactions. The first (Type I reaction) involves an initial electron transfer to ^3^O_2_ and the consequent formation of superoxide anion (O_2_^−^), from which hydrogen peroxide (H_2_O_2_) arises. This, in turn, can eventually generate the highly reactive and toxic hydroxyl radical (OH•). These species are collectively called reactive oxygen species (ROS). The second (Type II reaction) entails energy transfer from ^3^PS* to ^3^O_2_ with the production of another ROS, highly toxic singlet oxygen (^1^O_2_), which is the key species in PDT [2,3]. The local irradiation of a tumor after the systemic administration of PS can, therefore, lead to a burst of cytotoxic ROS in a confined area, triggering cell death with high spatial accuracy, however, the success of PDT is strictly related to the presence of O_2_, which is necessary for the production of ROS. Solid tumors have poor blood supply in their bulk and consequently have a low O_2_ concentration. This is the reason for the scant efficacy of PDT against these kinds of malignancies. A few strategies have recently been proposed to overcome this problem [4]. In this frame, one of the most appealing approaches is based on the use of nitric oxide (NO). NO is an endogenous messenger that is ubiquitous in mammalian tissues and cells. This ephemeral free radical, which has a half-life of ca. 5 s and diffusion radius of ca. 200 µm in tissues, is involved not only in maintaining and regulating important physiological processes, such as cell proliferation, mitochondrial oxidative metabolism, antimicrobial defences, and vasodilation, but also plays crucial roles in an extensive number of different diseases, including cancer [5,6]. It has been shown that low (nM) NO concentrations promote cancer growth while high (μM) concentrations are toxic and reduce cancer progression [7,8]. Other factors, besides concentration, can influence the effects of NO on tumor growth. These factors include the duration of NO exposure and cellular sensitivity in particular [7,9]. The toxic effects of NO can be both direct and indirect. The former is related to the capacity of NO to react with the transition metals in some biomolecules that are essential for cellular life (e.g., the iron-sulfur centers in proteins and iron-containing enzymes). The indirect effects are caused by the ability of NO to react with O_2_ or O_2_^−^•, affording reactive nitrogen oxide species (RNOS) that can oxidize, nitrate and nitrosate a variety of biological targets, such as ribonucleotide reductase and proteins that contain iron-sulfur clusters, such as aconitase and the components of mitochondrial respiration complexes [7,8]. Peroxynitrite (ONOO^−^) is one of these types of RNOS and is highly toxic. It is formed in the rapid reaction of NO with O_2_^−^• (k = 6, 7 × 10^9^ M^−1^ s^−1^) and is both a potent oxidant and nitrating agent. In the physiological environment, ONOO^−^ gives rise to a pair of OH• and NO_2_• radicals, via homolyses, that can combine to yield nitrate. In addition, it can interact with carbon dioxide (CO_2_), to produce the carbonate radical (CO_3_^−^•), which is another strong oxidant [10].

Therefore, it is not surprising that NO donor-based therapy has recently received particular attention as a potential clinical treatment for cancer. It is centred on the use of NO donors, namely products that can release NO under physiological conditions [11,12]. However, the main limitation of this approach is the lack of precise spatial-temporal control of the NO amount that is to be released which, as described above, is fundamental for a positive therapeutic outcome. For these reasons, there is currently great interest in compounds that are able to release NO under the action of light, namely NO photodonors (NOPDs). In these structures, NO is “caged”, by a covalent bond, into a photoactivatable scaffold that harnesses the excitation energy to break bonds and, consequently, either liberate NO [13,14,15,16,17,18,19,20] or compounds that, in turn, spontaneously release it [21,22,23,24,25,26]. In these cases, NO release is limited to the irradiated area and fine temporal control of its release is achieved by simply tuning the duration and intensity of the irradiation. NOPDs are especially appealing in the field of light-controlled NO anticancer photodynamic therapy (NOPDT) as it may be a means to overcome the limitations of PDT in hypoxic tumors [4]. The combination of PS and NOPDT is an ideal, innovative and unconventional methodology thanks to the important properties that ^1^O_2_ and NO share which include: (i) no resistance being developed towards these species, (ii) multitarget activity, and (iii) confinement of their action to a restricted region of space due to their short lifetimes and diffusion radius [27].

We have recently reported compound **1** (Scheme 1), in which a boron dipyrromethene (BODIPY) scaffold is covalently joined to cupferron, a C-diazeniumdiolate that is able to spontaneously release NO [26]. Under the action of biocompatible green light, **1** affords NO following the release of the cupferron moiety. In order to pursue the recent idea of incorporating a NOPD and a PS within the same molecular skeleton [28], we have devised the 2,6-diiododerivative, **2** (Scheme 1).

In this case, the presence of the two iodine substituents is expected to encourage the population of the triplet state, which is fundamental for ROS generation via type I and type II mechanisms (vide supra) without, hopefully, hampering NO photorelease properties. In this paper, we report the synthesis, and the photochemical and photophysical characterization of compound **2** and the validation of the cytotoxicity, against human melanoma A375 and SKMEL28 cell lines, of this compound, the non-iodinate analogue, **1**, and the 8-OCH_3_ substituted BODIPYs, **3** and **4**, which were chosen as suitable model compounds for **1** and **2**, respectively. The nature of the reactive species involved in the antiproliferative effects of these compounds is discussed with reference to a detailed study of their photochemical profiles.

## 2. Materials and Methods 

### 2.1. Materials

All reagents (Sigma-Aldrich) were of high commercial grade and were used without further purification. All solvents used (from Carlo Erba) were of spectrophotometric grade. 

### 2.2. Synthetic Procedures

All reactions involving air-sensitive reagents were performed under nitrogen in oven-dried glassware using the syringe-septum cap technique. All solvents were purified and degassed before use. Chromatographic separation was achieved under pressure using Merck silica gel 60 and flash-column techniques. Reactions were monitored by thin-layer chromatography (TLC) on 0.25 mm silica-gel-coated aluminium plates (Merck 60 F254) (Merck KGaA, Darmstadt, Germany) using UV light (254 nm) as the visualising agent. Unless otherwise specified, all reagents were used as received without further purification. Compounds **1** and **3** were synthesised according to the previously reported procedure [26,29].

#### 2.2.1. Synthesis of 2-((4,4-difluoro-2,6-diiodo-1,3,5,7-tetramethyl-4-bora-3a,4a-diaza-s-indacen-8-yl) methoxy)-1-phenyldiazene oxide 2

A stirred solution of compound **1** (40 mg, 0.10 mmol) in EtOH (35 mL) was treated with iodine (63.5 mg, 0.25 mmol) and then with iodic acid (42.1 mg, 0.25 mmol) in 1 mL of water. The reaction mixture was stirred at 80 °C for 3 h. Once this was completed, as shown by TLC analysis, the mixture was cooled to room temperature and the solvent was removed under reduced pressure. The residue was dissolved in dichloromethane (DCM), and then washed with water (2 × 20 mL) and brine (2 × 20 mL). The combined organic layers were dried over sodium sulphate. Purification via silica gel chromatography, eluting with 1/1 petroleum ether/DCM, gave **2** as a dark red crystalline solid (51 mg, 78%). M.p. = 195.7–196.8 °C (with dec.); ^1^H NMR (300 MHz, CDCl_3_): δ 7.91 (d, J = 7.2 Hz, 2H), 7.58–7.44 (m, 3H), 5.70 (s, 2H), 2.64 (s, 6H), 2.56 (s, 6H); ^13^C NMR (75 MHz, CDCl_3_): δ 158.4, 144.0, 143.2, 132.9, 131.9, 131.6, 129.3, 121.3, 87.7, 66.6, 18.8, 16.6; ESI-MS [M + Na]^+^: m/z 673.2; ESI-HRMS(+): m/z 672.95491 [M + Na]^+^, C_20_H_19_BF_2_I_2_N_4_O_2_Na requires 672.95506; HPLC purity >95% (CH_3_CN/H_2_O TFA 0.1% 90:10 (v/v), flow = 1.0 mL/min, t_R_ = 8.7 min) at 226, 254 and 550 nm.

#### 2.2.2. Synthesis of 4,4-difluoro-2,6-diiodo-8-methoxymethyl-1,3,5,7-tetramethyl-4-bora-3a,4a-diaza-s-indacene **4**

A solution of compound **3** (0.230 mmol) in EtOH (30 mL) was treated with iodine (0.570 mmol) and then with iodic acid (0.570 mmol) in 2 mL of water. The reaction mixture was stirred for 2 h at r.t., and the solvent was removed under reduced pressure. The residue was dissolved in EtOAc, and then washed with water (2 × 20 mL) and brine (2 × 20 mL). The combined organic layers were dried over sodium sulphate. Purification by silica gel chromatography, eluting with 95/5 petroleum ether/EtOAc, gave the title product, **4**, as a dark red solid (96 mg, 77%). M.p. = 186.4–187.0 °C (with dec.); ^1^H NMR (300 MHz, CDCl_3_): δ 4.54 (s, 2H), 3.48 (s, 3H), 2.61 (s, 6H), 2.46 (s, 6H); ^13^C NMR (75 MHz, CDCl_3_): δ 157.4, 143.9, 135.3, 133.1, 86.8, 65.6, 58.9, 17.8, 16.4; ESI-MS [M-H]^−^: m/z 543.2; HPLC purity >95% (CH_3_CN/H_2_O TFA 0.1% 90:10 (v/v), flow = 1.0 mL/min, t_R_ = 8.7 min) at 226, 254 and 550 nm.

### 2.3. Instrumentation

^1^H NMR and ^13^C NMR spectra were recorded on a Bruker Avance 300 (Bruker, Billerica, MA, USA) at room temperature at 300 and 75 MHz, respectively, and calibrated using SiMe_4_ as the internal reference. Chemical shifts (δ) are given in parts per million (ppm) and coupling constants (J) in Hertz (Hz). The following abbreviations are used to designate the multiplicities: s = singlet, d = doublet, t = triplet, q = quartet, m = multiplet, and br = broad. Low-resolution mass spectra were recorded on a Micromass Quattro microTM API (Waters Corporation, Milford, MA, USA). High-resolution mass spectra were recorded on a Bruker Bio Apex Fourier transform ion cyclotron resonance (FT-ICR) mass spectrometer equipped with an Apollo I ESI source, a 4.7 T superconducting magnet, and a cylindrical infinity cell (Bruker Daltonics, Billerica, MA, USA). HPLC analyses were performed using a HP 1200 chromatograph system (Agilent Technologies, Palo Alto, CA, USA) equipped with a quaternary pump (model G1311A, Agilent Technologies, Palo Alto, CA, USA), a membrane degasser (G1322A, Agilent Technologies, Palo Alto, CA, USA), and a multiple wavelength UV detector (MWD, model G1365D, Agilent Technologies, Palo Alto, CA, USA) integrated into the HP1200 (Agilent Technologies, Palo Alto, CA, USA) system. Data analysis was performed using a HP ChemStation system (Agilent Technologies). The sample was eluted in a Merck LiChrospher C18 end-capped column (250 × 4.6 mm ID, 5 µm) (Merck KGaA, Darmstadt, Germany). The injection volume was 20 μL (Rheodyne, Cotati, CA, USA). The mobile phase consisted of acetonitrile 0.1% TFA (solvent A) and 0.1% TFA (solvent B) at flow rate = 1.0 mL/min. For the photoproduct distribution studies, irradiation was performed using a white LED 90W lamp (Led Engin, San Jose, CA, USA) equipped with a green filter BP-540 80HT (transmission range 515–565 nm, T >90%, Schneider, Kreuznach, Germany) as the light source. Irradiance was measured using a HD2302.0 Delta Ohm lightmeter (Delta Ohm, Caselle di Selvazzano (PD), Italy) equipped with a Delta Ohm LP471RAD light probe (Delta Ohm, Caselle di Selvazzano (PD), Italy).

UV-Vis absorption and fluorescence emission spectra were recorded on a JascoV-560 spectrophotometer (Jasco, Easton, MD, USA) and a Spex Fluorolog-2 (mod. F-111) spectrofluorimeter (Horiba, Kyoto, Japan), respectively, in air-equilibrated solutions, using quartz cells with a path length of 1 cm. Fluorescence lifetimes were recorded with the same fluorimeter, which was equipped with a Time-Correlated Single Photon Counting (TCSPC) Triple Illuminator (Horiba, Kyoto, Japan). The samples were irradiated by a pulsed diode excitation source (Nanoled) at 455 nm and the decays were monitored at 500 nm. The system allowed fluorescence lifetimes to be measured from 200 ps. The multiexponential fit of the fluorescence decay was obtained using the following equation:I(t) = Σα_i_exp(^−t/τ_i_^)

Absorption spectral changes were monitored by irradiating the sample in a thermostated quartz cell (1 cm path length, 3 mL capacity) under gentle stirring, using a continuum laser with λexc = 532 nm, ca. 50 mW and a beam diameter of ca. 1.5 mm.

The direct monitoring of NO release in solution was performed via amperometric detection (World Precision Instruments, Sarasota, FL, USA), with a ISO-NO meter, equipped with a data acquisition system, and was based on the direct amperometric detection of NO with a short response time (<5 s) in a 1 nM to 20 µM sensitivity range. The analogue signal was digitalized using a four-channel recording system and transferred to a PC. The sensor was accurately calibrated by mixing standard solutions of NaNO_2_ with 0.1 M H_2_SO_4_ and 0.1 M KI, according to the reaction:4H^+^ + 2I^−^ + 2NO_2_^−^ → 2H_2_O + 2NO + I_2_

Irradiation was performed in a thermostated quartz cell (1 cm path length, 3 mL capacity) using the continuum laser at λexc = 532 nm. NO measurements were carried out under stirring with the electrode positioned outside the light path in order to avoid NO signal artefacts that could be caused by photoelectric interference with the ISO-NO electrode.

^1^O_2_ emission was registered using a Fluorolog-2 (Model, F111) spectrofluorimeter that was equipped with a NIR-sensitive liquid nitrogen cooled photomultiplier, which excited the air-equilibrated samples with a CW laser at 532 nm.

#### Laser Flash Photolysis

All solutions were excited with the second harmonic of a Nd-YAG Continuum Surelite II-10 laser (532 nm, 6 ns FWHM) (Continuum, Santa Clara, CA, USA), using quartz cells with path length 1.0 cm. The excited solutions were analyzed using Luzchem Research mLFP-111 apparatus (Luzchem Research Inc., Gloucester, ON, Canada) with an orthogonal pump and probe configuration. The probe source was a ceramic xenon lamp coupled to quartz fibre optic cables. The laser pulse and the mLFP-111 system were synchronized using a Tektronix TDS 3032 digitizer (Tektronix, Beaverton, OR, USA), which was operated in pretrigger mode. The signals from a compact Hamamatsu photomultiplier (Hamamatsu Photonics K.K., Hamamatsu, Japan) were initially captured by the digitizer and then transferred to a personal computer, which was controlled by Luzchem Research software (Luzchem Research Inc., Gloucester, ON, Canada) operating in the National Instruments LabView 5.1 (National Instruments, Austin, TX, USA) environment. The solutions were deoxygenated via bubbling with a vigorous and constant flux of pure nitrogen (previously saturated with solvent). The solution temperature was 295 ± 2 K. The energy of the laser pulse was measured at each shot with a SPHD25 Scientech pyroelectric meter (Scientech Inc., Boulder, CO, USA).

### 2.4. Biological Experiments

#### 2.4.1. Cell Cultures

Culture media were supplied by Invitrogen Life Technologies (Carlsbad, CA, USA), and plasticware for the cell cultures was obtained from Falcon (Becton Dickinson, Franklin Lakes, NJ, USA). The protein content of the cell monolayers was assessed using the BCA kit from Sigma Chemical Co. (St. Louis, MO, USA). Unless otherwise specified, all reagents were from Sigma Chemical Co. The compounds were dissolved in dimethyl sulfoxide (DMSO). The stock solution was diluted in culture medium to reach the final concentration. The concentration of DMSO in the culture medium under each experimental condition was less than 0.1%.

Human melanoma A375 and SKMEL28 cells (ATCC, Manassas, VA, USA) were cultured in DMEM medium that was supplemented with 10% fetal bovine serum and 1% penicillin-streptomycin. Cell cultures were maintained in a humidified atmosphere at 37 °C and 5% CO_2_. When indicated, cells were incubated for 4 h with the compounds and then exposed for 30 min to the light emitted by a white LED 90W lamp with an irradiance of 1.75 mW/cm^2^ and a bandpass filter of 540 nm ± 40 nm (IFG BP 540-80 HT), in DMEM medium without phenol red at room temperature. Non-irradiated cells were maintained in a dark room for 30 min in DMEM medium without phenol red at room temperature. After this period, cells were left for 19.5 h in the incubator before the experimental procedures, described below, were performed.

#### 2.4.2. NO Photorelease in Cells

After the cells were incubated and irradiated, as described in the “Cell” section, 1 mL of cell supernatant was collected and centrifuged for 10 min at 13,000× *g*. The presence of nitrite in the reaction mixture was then determined using the Griess assay [30], and a Synergy HT microplate reader (Bio-Tek Instruments, Winooski, VT, USA).

#### 2.4.3. Cytotoxicity

The cytotoxic effect of the compounds was measured, using a Synergy HT microplate reader (Bio-Tek Instruments, Winooski, VT, USA), as previously described [31], as the leakage of lactate dehydrogenase (LDH) into the extracellular medium. Both intracellular and extracellular LDH were measured, and then extracellular LDH activity (LDH out) was calculated as a percentage of the total (intracellular + extracellular) LDH activity (LDH tot) in the dish. For the calculation of IC_50_, i.e., the concentration of each compound that decreased the cell viability by 50%, cells were incubated with either 0, 0.1, 0.25, 0.5, 1, 2.5, 5, 10, 25, 50, 75, or 100 μM of the single compounds, and were either maintained in the dark or irradiated for 30 min. Viable cells were measured after 24 h using the ATPlite kit (PerkinElmer, Waltham, MA, USA). Chemiluminescence was read using a Synergy HT microplate reader. The mean relative luminescence units (RLUs) of the untreated cells was considered to be 100%, and the RLUs of the other experimental conditions were expressed as percentage of viable cells vs. untreated cells. IC_50_ was calculated using GraphPad Prism (v.6.01) software.

#### 2.4.4. Statistical Analysis

All the data in the text and figures are provided as means ± SD. The results were analysed via a one-way analysis of variance (ANOVA) and Tukey’s test. *p* < 0.05 was considered significant.

## 3. Results and Discussion

### 3.1. Photochemical and Spectroscopic Properties

BODIPY derivative **2** is highly soluble in the H_2_O:MeOH (20:80 v/v) solution, and its absorption features are dominated by the BODIPY chromophore in the visible region (Figure 1A). This compound is very stable under these conditions at room temperature in the dark for several days. By contrast, clear photodegradation is observed upon irradiation with green light at 532 nm (Figure 1). It is worth noting that neither the photolysis nor rate profile were affected by the presence of oxygen (see inset Figure 1A), suggesting that the primary photodegradation pathway proceeds from a short-lived excited state (i.e., the lowest singlet). The HPLC analysis of the irradiated solution revealed the presence of two solvent substitution products that lack the cupferron moiety, accompanied by nitrosobenzene (Figure 1B). These findings account for a photodegradation mechanism that is similar to the one that has already been observed for the non-iodinate analogue **1** [26], and that involves the heterolytic rupture of the C-O bond, with the consequent formation of a BODIPY-methyl carbocation via solvent substitution (see Scheme 1). Furthermore, the presence of nitrosobenzene as a photolysis product suggests that NO is quickly lost from the photodecaged cupferron. NO release was monitored via its real-time detection by an ultrasensitive NO electrode, which directly detects NO concentration using an amperometric technique. The results illustrated in Figure 1C provide clear evidence that compound **2** is stable in the dark, but that it generates NO under irradiation with Vis green light.

As expected, the presence of the iodine substituents in compound **2** dramatically amplified the ISC process leading to a drastic reduction in fluorescence emission and lifetime (more than two orders of magnitude compared with **1**) and the effective population of the lowest excited triplet state, in contrast to non-iodinate **1**. Laser flash photolysis with nanosecond time resolution is a powerful tool with which to examine the spectroscopic and kinetic features of excited triplets of many PSs, since these transient species exhibit very intense absorption in the visible region, and lifetimes in the order of microseconds. Figure 2A shows the transient absorption spectrum of compound **2** observed 0.1 µs after the laser pulse. It consists of two bands, around 440 nm and 650 nm, and bleaching in the region that corresponds to the ground state absorption of the BODIPY chromophore, which is in line with the typical triplet state absorption of BODIPY derivatives [32]. The triplet state of **2** decays monoexponentially (inset Figure 2A) with a lifetime in the microsecond time regime in N_2_-saturated solution and is quenched by molecular oxygen with a diffusional quenching constant. Accordingly, a negligible triplet signal was observed in the optically matched solution of **1** (see Figure 2A). As outlined above, triplet quenching by molecular oxygen can lead to either ^1^O_2_ or O_2_^−^ production, according to the type of mechanism. ^1^O_2_ was detected directly by measuring its typical phosphorescence in the near-IR spectral window, upon excitation with green light. Figure 2B shows that the typical luminescence signal of ^1^O_2_, with maximum at ca. 1270 nm, is only one observed in the case of compound **2**.

The spectroscopic and photochemical scenario described above, therefore, demonstrates that the cupferron decaging, and subsequent NO release, occurs from the excited singlet state and that ^1^O_2_ is the main quenching product from the lowest excited triplet state, according to Scheme 2.

### 3.2. Biological Experiments

#### 3.2.1. NO Photorelease in Cells

The NO photorelease demonstrated by **1** and **2** was also investigated in A375 and SKMEL28, two melanoma cell lines. After 4 h of incubation with the products at three different concentrations, the cells were irradiated for 30 min with the light emitted by a white LED 90 W lamp with an irradiance of 1.75 mW/cm^2^ and a bandpass filter 540 nm ± 40 nm. NO was measured as nitrite (its main oxidation product) as determined in the supernatants by Griess assay. Similar experiments were carried out on model compounds **3** and **4** which, unsurprisingly, failed to show any nitrite generation (data not shown). The data reported in Figure 3 clearly demonstrate that products **1** and **2** were able to produce nitrite when irradiated in cells. In particular, they afforded significant amounts of nitrite in the two cell lines at 2.5 to 5 µM concentrations, i.e., in a concentration range that is compatible with the possible cytotoxic and antitumor effects of NO-derived reactive species [7,8,9]. The amount of nitrite produced by **1** and **2** was similar in the two cell lines. Individual cell lines can differ somewhat in how enzymes metabolize NO and its stable derivatives; superoxide dismutase neutralises O_2_^−^ and limits the generation of ONOO^−^, while nitrate reductase increases the conversion of nitrate into nitrite and amplifies the levels of nitrite that are detected using the Griess method. The cell independency shown in nitrite production, here, strongly suggests that NO generation is dependent on the photochemical properties of the compounds. **1** is a more efficient nitrite producer than **2** in the 2.5–5 µM concentration range. Since we have ascertained that NO photorelease takes place from the lowest excited singlet state of both **1** and **2**, the lower efficiency found for **2** reflects the effective ISC that occurs in this compound, which competes with the cupferron decaging and therefore impacts upon the NO produced.

#### 3.2.2. Cytotoxicity against Human Melanoma A375 and SKMEL28 Cells

A375 and SKMEL28 cells were incubated with 1, 2.5, and 5 µM of compounds **1** and **2** and were then either left in the dark or irradiated for 30 min with green light. Cytotoxicity was evaluated 24 h after the completion of irradiation by measuring the release of LDH, a sensitive index of membrane damage due to oxidative [31] and nitrosative stress [33], which correlates well with cell death [34]. The results shown in Figure 4 demonstrate that both products are nontoxic in the dark, but that they show a good profile of activity, which is finely dependent on concentration, under irradiation. While the cytotoxicity of **2** is similar in the two cell lines, **1** was more toxic in A375 than in SKMEL28 cells. According to NO photogeneration and the lack of ^1^O_2_ photosensitisation, the toxicity observed in the case of **1** is most likely principally due to NO toxicity and not mediated by ONOO^−^. The differential cytotoxicity of **1**, in particular in the 1–2.5 µM range, in the A375 and SKMEL28 cells may be due to the fact that the former are more dependent on efficient mitochondrial oxidative respiration than the latter; inhibiting mitochondrial respiration with KCN reduces cell viability in A375 cells more than in SKMEL28 cells [35]. Since NO is a proven inhibitor of complexes I, III, and IV [7,8,9], as well as of aconitase, in this range of concentration, the cytotoxicity caused by **1** is most likely due to the impairing of mitochondrial respiration elicited by the released NO.

Moreover, the activity profile observed for compound **2** against both cell lines shows higher cytotoxicity than **1**. In order to establish whether this higher cytotoxicity is due to ^1^O_2_, NO or the combined action of the two species, cytotoxicity experiments were also performed with the model compounds **3** and **4**. The results reported in Figure 5 show that **3** fails to show toxicity in the dark and upon irradiation. This is not surprising as this compound does not produce NO and does not generate ^1^O_2_. By contrast, compound **4** shows toxicity under green light irradiation at concentrations higher than 2 µM. As we found that **4** generates ^1^O_2_ with similar efficiency to **2**, the photodynamic action observed can be exclusively attributed to this transient species. A comparison of the data in Figure 4 and Figure 5 provides quite a clear scenario of the biological effects. In fact, compound **2** already exhibits relevant levels of phototoxicity at a concentration of 1 µM in both cell lines, unlike model compound **4**, which is inactive at the same concentration. These data, together with the phototoxicity observed for compound **1**, which exclusively produces NO, suggest that the phototoxic effects mediated by **2** are clearly a result of the combined action of ^1^O_2_ and NO. We can infer that the higher cytotoxicity that **2** shows, in particular in the 1–2 µM concentration range, as compared with **4** at these concentrations can be explained by the different reactive species generated by the compounds upon irradiation, i.e., ^1^O_2_ and NO from **2**, only ^1^O_2_ from **4**. This comparison allows us to separate the proportions of the cytotoxic effects of NO and those of ^1^O_2_.

The results obtained from the LDH release assays were further confirmed by the calculation of IC_50_ values (Table 1). The IC_50_ values for all of the compounds were higher than the maximum concentration tested and compatible with a good solution of the compounds, i.e., 100 μM, indicating that the compounds did not exert peculiar toxicity if not irradiated. Upon irradiation, the IC_50_ of **1**, **2**, and **4** was significantly lower. In particular, the IC_50_ of **2** was lower than the toxicity of **1**, which is in line with the ability of **2** to release two toxic species, i.e., NO and ^1^O_2_, and the ability of **1** to release NO only. The IC_50_ of the respective model compounds, **3** and **4**, were higher. The IC_50_ of **3** was similar to that measured in the dark, since this compound does not produce either NO or ^1^O_2_ upon irradiation. The IC_50_ of **4**, which generates ^1^O_2_ similarly to **2**, but not NO, is lower than the IC_50_ of **2** upon irradiation. These data confirm that the maximal phototoxicity is produced by the simultaneous release of RNS and ROS, as occurs in compound **2**.

## 4. Conclusions

In conclusion, we have reported a new BODIPY derivative, **2**, that is able to release NO and ^1^O_2_ under the action of green light. The photogeneration of these transient cytotoxic species has been unambiguously demonstrated by their direct detection. With the aid of analogue derivative **1**, which exclusively photogenerates NO, and two suitable model compounds based on the same BODIPY core, we have provided evidence that the in vitro cytotoxic activity observed upon green light irradiation of **2** against human melanoma cell lines is due to the combined action of NO and ^1^O_2_, whereas only NO is involved in the antiproliferative effects induced by **1**. Our results clearly demonstrate that the possibility of releasing both NO and ^1^O_2_ in a spatially and temporally controlled manner is the most powerful tool for killing tumor cells, while sparing non-transformed, non-irradiated healthy tissues. These data are especially promising as they have been obtained against melanoma cells, which are known for their high resistance to chemotherapy and targeted therapy [36]. Interestingly, NO photogeneration is not affected by the presence of oxygen. These findings, therefore, provide further confirmation that strategic approaches that are based on the photoregulated release of NO and its combination with ROS are very appealing as potential tools for the innovative treatment of tumors. Moreover, our compounds are active in vitro in low micromolar concentrations, like most chemotherapeutic drugs. The BODIPY derivatives used in the present study do not show preferential accumulation within a specific organelle, as demonstrated previously by [16]. However, the BODIPY scaffold can be chemically conjugated with mitochondria, endoplasmic reticulum, and lysosome-vectorising moieties. Work on this topic is currently ongoing in our group. Using this approach, we plan to concentrate the simultaneous release of ROS and RNS within a single compartment, and thus produce higher selective damage. Indeed, it is known that RNS exerts antitumor effects by inhibiting mitochondrial metabolism, inducing ER stress, and regulating lysosomal functions when released at micromolar concentrations [11]. The targeted release of RNS and ROS within a specific compartment will boost the cytotoxic potential of our hybrids. Our present work is a proof of concept that demonstrates the opportunity provided by a technique that can achieve the same antitumor efficacy of classical chemotherapeutic drugs, avoid the problem of the onset of chemoresistance and undesired side effects, while also exploiting the intratumor destruction produced by oxidative and nitrosative damage upon irradiation with visible light.

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
