# Peer review of "Combination of PDT and NOPDT with a Tailored BODIPY Derivative"

_antioxidants, 2019, doi:10.3390/antiox8110531_

Round 1

Reviewer 1 Report

Recommendation: Acceptable after major revision

This manuscript emphasizes the newly developed oxygen independent NO-anticancer photodynamic therapy (NOPDT) based on BODIPY backbone because most of the tumor has innate hypoxia condition, which limits the efficient cancer therapy. The proposed organic photosensitizers utilized BODIPY core unit which were substituted with Cupferron and iodine for NO and reactive oxygen species (ROS) generation respectively. Based on this photosensitizers, controllable NO photo-release from compound 1 and compound 2 is shown. In addition, combination of previous PDT and NOPDT improves therapeutic efficacy in cancer cells. There are some concerns before publication. This manuscript will be acceptable after the comments are properly addressed.

Figure 2 (a) inset should provide χ2 value for its single exponential fitting.

The big advantage of NOPDT is efficient cancer therapy under hypoxia condition, however, the authors did not provide proper evidence of the efficacy of compound 1 and compound 2 under hypoxia condition.

-> In hypoxia condition, the author should be compare the respective therapeutic efficacy of only NOPDT (with compound 1), only PDT (with compound 4) and combination of PDT and NOPDT (compound 2)

In page 9 line 299~301, RNS and ROS generation inevitably compete to each other in suggested BODIPY compounds. I believe the photosensitization process may not be confined to intramolecular dynamics of single-molecular compound Rather, Scheme 2 should add another process of photolysis compound: the separate process accompanying with the second photoexcitation of ground state ‘compound 2 fragment’ (possibly compound 4) generating ROS. Relatively long light exposure time of 30min also supports the other possibility of ROS generation from the fragment of compound 2.

I wonder about ‘what is target organelle of each BODIPY compound’.

IC50 values for quantification of therapeutic efficacy should be provided.

Author Response

This manuscript emphasizes the newly developed oxygen independent NO-anticancer photodynamic therapy (NOPDT) based on BODIPY backbone because most of the tumor has innate hypoxia condition, which limits the efficient cancer therapy. The proposed organic photosensitizers utilized BODIPY core unit which were substituted with Cupferron and iodine for NO and reactive oxygen species (ROS) generation respectively. Based on this photosensitizers, controllable NO photo-release from compound 1 and compound 2 is shown. In addition, combination of previous PDT and NOPDT improves therapeutic efficacy in cancer cells. There are some concerns before publication. This manuscript will be acceptable after the comments are properly addressed.

1) Figure 2 (a) inset should provide χ2 value for its single exponential fitting.

We inserted the R2 value at the end of Figure 2A caption

2) The big advantage of NOPDT is efficient cancer therapy under hypoxia condition, however, the authors did not provide proper evidence of the efficacy of compound 1 and compound 2 under hypoxia condition. In hypoxia condition, the author should be compare the respective therapeutic efficacy of only NOPDT (with compound 1), only PDT (with compound 4) and combination of PDT and NOPDT (compound 2)

The efficacy of photodynamic therapy in hypoxic tumors is limited in hypoxic tumors, although some advance in this filed has been proposed [Li, X., et al. Angew. Chem. Int. Ed. 2018, 57, 11522-11531; ref. 4 of the manuscript]. We hypothesize that NOPDT may overcome the limitation of PDT in hypoxic tumors, although we fully agree with the Reviewer’s comment that a proof of the efficacy of compound 1, 2 and 4 in hypoxia will strengthen our work. At the moment, however, this experimental set is not feasible because we can maintain culture cells before and after irradiation in hypoxic conditions in a dedicated incubator with O2 tension ranging from 20% to 1%, but we perform the irradiation step in normoxic conditions. Since also a 30 minutes exposure to normoxic conditions could be sufficient to alter the cellular responses in terms of metabolism, proliferation and susceptibility to damaging agents [Almendros, I., et al, Respir. Physiol. Neurobiol. 2013, 186, 303-307; Hunyor, I., and Cook, K.M., Am. J. Physiol. Regul. Integr. Comp. Physiol. 2018, 315, R669-R687; Martinez, C.A., et al., Int. J. Mol. Sci. 2019, 20, pii: E445], we feel not confident to present any data on cytotoxicity with this experimental setting, because they may be biased by the alternation of hypoxic and normoxic conditions. We plan to adopt a hypoxic chamber, where all the steps- cellular seeding and growth, irradiation, post-irradiation period before measuring the cytotoxicity can be performed in a stable hypoxic environment. These experiments will be object of a future work.

Knowing that we cannot stated that the data presented in this work demonstrate the advantage of NOPDT in hypoxia, we smoothened the Introduction (line 81) and the Conclusions (line 399) paragraph accordingly

3) In page 9 line 299~301, RNS and ROS generation inevitably compete to each other in suggested BODIPY compounds. I believe the photosensitization process may not be confined to intramolecular dynamics of single-molecular compound Rather, Scheme 2 should add another process of photolysis compound: the separate process accompanying with the second photoexcitation of ground state ‘compound 2 fragment’ (possibly compound 4) generating ROS. Relatively long light exposure time of 30min also supports the other possibility of ROS generation from the fragment of compound 2.

We agree with the referee’s comment regarding the potential involvement of the photoproduct 4 as singlet oxygen photosensitizer. However, Scheme 2 is shown to better highlight that ROS and RNS are simultaneously generated from the same photoprecursor (compound 2), as noted by their direct detection. By taking into account the efficiency of the photodegradation of compound 2 and the light intensity used for the 30 min irradiation in the biological experiments, the contribution of photoproduct 4 to the cell viability observed can be considered negligible. 

3) I wonder about ‘what is target organelle of each BODIPY compound’.

The BODIPY-derivatives used in the present study show a homogeneous intracellular accumulation, without showing a preferential accumulation within a specific organelle, as demonstrated by previous works of our group [C. Parisi, et al., Chem. Eur. J. 2019, 47, 11080-11084; ref. 16 of the manuscript]. However, BODIPY scaffold may be chemically conjugated with mitochondria, endoplasmic reticulum or lysosome-vectorising moieties. This work is currently ongoing in our group. By this approach we plan to concentrate the simultaneous release of ROS and RNS within one compartment, producing higher selective damages. Indeed, it is know that RNS exert anti-tumor effects by inhibiting mitochondrial metabolism, inducing ER stress, and regulating lysosomal functions when released at micromolar concentrations [Huerta, S., et al., Int. J. Oncol. 2008, 33, 909-927; ref. 11 of the manuscript]. The targeted release of RNS and ROS within a specific compartment will boost the cytotoxic potential of our hybrids.

We added a comment on this point in the Conclusion paragraph (line 402).

4) IC50 values for quantification of therapeutic efficacy should be provided.

We added the IC50 values in the new Table 1. The results of IC50 confirmed the data of the LDH release assays. Indeed, the IC50 for all compounds was similar in the dark and fall in the high micromolar range, indicating that the compounds did not exert peculiar toxicity if not irradiated. Upon irradiation, the IC50 of  1, 2 and 4 was significantly lower. In particular, the IC50 of 2 was lower than the toxicity of 4, in line with the ability of 2 of releasing two toxic species, i.e. NO and 1O2, and the ability of 1 of releasing only NO. The IC50 of the respective model compounds, 3 and 4, was higher. The IC50 of 3 was similar to that measured in the dark, since this compound does not produce NO nor 1O2 upon irradiation. The IC50 of 4, that generates 1O2 similarly to 2 but not NO, is lower than the IC50 of 2 upon irradiation. These data confirm that the maximal phototoxicity is produced by the simultaneous release of RNS and ROS, as it occurs for compound 2.

We modified the Results section (line 368) and the Materials and Methods section (line 230) accordingly.

Reviewer 2 Report

This is a well written and thorough manuscript describing the synthesis of a BODIPY derivative which is both a photosensitizer and a nitric oxide photogenerator. Both reactive species (superoxide and nitric oxide) have been directly observed after 532 nm, green, laser excitation. Additionally, cell line studies have shown in vitro cytotoxic activity for this BODIPY derivative.

I don't believe that this manuscript fits very well with this journal due to not discussing any antioxidant reactions. However, it certainly does fit the remit of the special issue Free Radical Research in Cancer and I would, therefore, recommend it for publication as it is. It would be good, in my opinion, to do a quick English and spelling check of the manuscript. For example, the sentence starting on line 364 would read better as 'Therefore, these findings provide a further confirmation that ...ROS are very appealing as potential tools ...' I also think it would be helpful to include a reference relating to the sentence on lines 52-54, about low and high concentrations of NO. However, I would still be happy to accept it in its present form.

Author Response

We thanks the Reviewer for the appreciation of our manuscript.

Following her/his advice, we add two references supporting the sentence on line 52-54 (line 56 in the revised version), i.e. reference 7 and reference 9 of the old version (now become reference 8).

We carefully revised the spelling and the grammar of the whole manuscript.

Reviewer 3 Report

This work by Salvatore Sortino et al. described “Combination of PDT and NOPDT with a Tailored 3 BODIPY Derivative”. The paper is complete and well structured. The description of experimental procedure is detailed and results critically analyzed and commented. Particularly interesting are results obtained in the biological experiments as a proof of concept showing the possibility to achieve the same anti-tumor efficacy obtained with classical chemotherapeutic drugs, avoiding the problem of the onset of chemoresistance and undesired side effects, but exploiting the intratumor damage produced by oxidative/nitrosative stress upon irradiation with visible light.

I think the manuscript can be accepted in this form and it needs no revision.

Author Response

This work by Salvatore Sortino et al. described “Combination of PDT and NOPDT with a Tailored 3 BODIPY Derivative”. The paper is complete and well structured. The description of experimental procedure is detailed and results critically analyzed and commented. Particularly interesting are results obtained in the biological experiments as a proof of concept showing the possibility to achieve the same anti-tumor efficacy obtained with classical chemotherapeutic drugs, avoiding the problem of the onset of chemoresistance and undesired side effects, but exploiting the intratumor damage produced by oxidative/nitrosative stress upon irradiation with visible light.

I think the manuscript can be accepted in this form and it needs no revision.

We thanks the Reviewer for the appreciation of our manuscript.

Round 2

Reviewer 1 Report

The modified manuscript is suitable for publication in this journal.